# Design and Validation of a Brain-Controlled Hip Exoskeleton for Assisted Gait Rehabilitation Training

**DOI:** 10.3390/mi16121364

**Published:** 2025-11-29

**Authors:** Chengjun Wang, Biao Cheng, Qiang Tang, Renyuan Wu, Huanyu Li

**Affiliations:** 1School of Artificial Intelligence, Anhui University of Science and Technology, Huainan 232001, China; wangchengjun@ahpu.edu.cn (C.W.); a15029238138@163.com (H.L.); 2State Key Laboratory of Digital Intelligent Technology for Unmanned Coal Mining, Anhui University of Science and Technology, Huainan 232001, China; 3School of Artificial Intelligence, Anhui Polytechnic University, Wuhu 241000, China

**Keywords:** gait rehabilitation, brain-controlled, hip exoskeleton, kinematic redundancy, augmented reality and visual stimulation (AR-VS) paradigm, hybrid control

## Abstract

This study presents an integrated micro-system solution to address the challenges of gait instability in patients with impaired hip motor function. We developed a novel wearable hip exoskeleton, where a flexible support unit and a parallel drive mechanism achieve self-alignment with the biological hip joint to minimize parasitic forces. The system is driven by an active brain–computer interface (BCI) that synergizes an augmented reality visual stimulation (AR-VS) paradigm for enhanced motor intent recognition with a high-performance decoding algorithm, all implemented on a real-time embedded processor. This integration of micro-sensors, control algorithms, and actuation enables the establishment of a gait phase-dependent hybrid controller that optimizes assistance. Online experiments demonstrated that the system assisted subjects in completing 10 gait cycles with an average task time of 37.94 s, a correlated instantaneous rate of 0.0428, and an effective output ratio of 82.17%. Compared to traditional models, the system achieved an 18.64% reduction in task time, a 28.31% decrease in instantaneous rate, and a 7.36% improvement in output ratio. This work demonstrates a significant advancement in intelligent micro-system platforms for human-centric rehabilitation robotics.

## 1. Introduction

Stroke, spinal cord injury, muscular dystrophy, and other neurological diseases or accidental injuries are easy to trigger in patients with impaired hip motor function and gait abnormality, forcing such patients to be in a state of physical limitation at all times, which seriously affects the daily life of patients [1,2,3]. Globally, stroke is the most common cause of long-term neurological deficits and severe functional disability, with approximately 76% of stroke survivors experiencing motor dysfunction, primarily characterized by limb weakness and joint stiffness [4,5]. Among them, hip joint motor dysfunction accounts for the highest rate. A large number of patients with impaired hip joints have problems with abnormal gait, such as Trendelenburg Gait, Antalgic Gait, and Flexion Contracture Gait, in the later stages of rehabilitation, and are prone to negative emotions such as depression and anxiety [6]. To prevent patients from developing secondary conditions, there is an urgent need to develop rehabilitation systems to help patients regain hip motor function and normal gait.

The initial hip exoskeleton designed by researchers was based on a wearable rigid exoskeleton, but during exoskeleton-assisted hip rehabilitation, it often left patients with walking difficulties, delayed movement, and a high sense of compression [7,8,9]. Considering the comfort of patients during training, it is important to first ensure that the exoskeleton has a low weight, has no delay in movement, and has unimpeded movement [10]. However, this is challenging with conventional rigid exoskeletons.

In response to the aforementioned challenges, wearable hip exoskeletons have been improved and primarily categorized into two main types based on their structure: rigid (anthropomorphic) exoskeletons and soft exoskeletons (exosuits) [11]. Considerable research efforts have been devoted to rigid exoskeletons. For instance, Pan Y.T. et al. [12] investigated the role of an anthropomorphic hip exoskeleton in gait rehabilitation assistance for stroke patients. Zhang T. et al. [13,14] proposed a lightweight wearable hip exoskeleton driven by a series elastic actuator with a dual-motor variable-speed drive, which is capable of providing an auxiliary torque of up to 60 N·m. A persistent challenge with these rigid designs, however, is the misalignment between the exoskeleton’s joints and the human biological hip joints during motion [15]. While solutions such as predicting the hip joint center for manual alignment or introducing kinematic redundancy with additional degrees of freedom have been proposed [16,17], they often result in bulky mechanisms, undermining the wearability of the devices. To overcome the mechanical constraints of rigid structures, research focus has significantly shifted to soft exoskeletons. Notable recent advances include reconfigurable systems like the multi-joint exomuscle developed by Ma et al. [18], which can be adapted to assist different lower-limb joints. The field features diverse actuation methods, as exemplified by the cable-driven exosuit for hip motion by Ye et al. [19], the soft pneumatic exoskeleton for rehabilitation by Miller-Jackson et al. [20], and the cable-driven exosuit for hip abduction assistance by Yang et al. [21]. A prevalent issue among the cable-driven system [19,21,22], however, is their reliance on Bowden cables for force transmission. This method imparts high tensile forces on the limb, generating substantial parasitic loads that can lead to harmful compression of the hip joint, thereby presenting a significant challenge for rehabilitation applications.

Wearable exoskeletons can be classified not only by structure but also by control method, primarily falling into passive and active types [23,24]. Representative examples include a multi-joint exoskeleton with active hip extension and passive ankle assistance [25], a dual-mode strategy emphasizing human–robot synergy in active training and movement guidance in passive training [26], and a neuro-fuzzy compensated PID controller for passive gait rehabilitation [27]. Typically, passive systems follow rule-based control with preset trajectories [28,29,30], but their repetitive nature can reduce patient engagement. In contrast, active systems first detect movement intention and then generate exoskeleton commands via algorithms [31,32,33], better accommodating patient volition and improving rehabilitation initiative. While both EMG and EEG signals can decode intention, EEG is more efficient due to earlier detection [34]. Among EEG paradigms, Steady-State Visual Evoked Potentials (SSVEP) has become the most prevalent, as it requires less user training and offers higher practicality compared to motor imagery [35,36,37,38].

This study introduces a brain-controlled hip exoskeleton for gait rehabilitation, featuring two degrees of freedom per leg (sagittal flexion/extension and coronal abduction/adduction). A parallel drive mechanism with a kinematically redundant flexible support unit ensures self-alignment with the hip joint, minimizing parasitic forces by directing assistance along the wearer’s instantaneous motion. Control is achieved through a dynamic augmented reality–visual stimulation (AR-VS) paradigm that integrates machine vision for real-time hip tracking and Steady-State Visual Evoked Potentials (SSVEP) for gaze-based command generation. System robustness is enhanced by a Filter Bank Canonical Correlation Analysis (FBCCA) decoding algorithm, while a gait phase-dependent hybrid controller optimizes actuation efficiency.

This integrated system, therefore, represents a convergence of advanced engineering domains. Its development hinges on the integration of micro-systems and intelligent devices, addressing core challenges in human–robot interaction through the synergy of micro-sensors, embedded processors, and micro-actuators.

This study makes the following key contributions:

(1) The design of a wearable hip exoskeleton that integrates a flexible support unit and a parallel drive mechanism. This self-aligning micro-mechatronic system minimizes parasitic forces and harmful hip compression by leveraging kinematic redundancy, enhancing wearability and comfort.

(2) The development of an integrated intelligent control system centered around an active BCI. This system synergizes an augmented reality–visual stimulation (AR-VS) paradigm with a high-performance decoding algorithm, implemented on a real-time embedded processor, to enable patient-autonomous, real-time control. Furthermore, a gait phase-dependent hybrid control strategy is proposed for efficient micro-actuator drive control.

This paper proceeds as follows: Section 2 presents the structural design and modeling analysis of the hip exoskeleton. Section 3 describes the control system for the active brain-controlled hip exoskeleton. Section 4 presents the experiments and results. Finally, Section 6 concludes the paper.

## 2. Structural Design and Modeling Analysis

### 2.1. Exoskeletal Structure

To more efficiently assist patients with impaired hip motor function to carry out gait rehabilitation training, this study designed a novel wearable hip exoskeleton based on the driving characteristics of the human biological hip joint, and a three-dimensional model of the exoskeleton is shown in Figure 1. The main components of the hip exoskeleton include a belt, two leg braces, four flexible struts, two parallel mechanisms, two drive units, and a control unit. Among them, the belt, leg braces, and flexible struts constitute the flexible support unit of the exoskeleton, as shown in Figure 2a; the parallel mechanism and the drive unit constitute the parallel drive mechanism. The exoskeleton has two degrees of freedom for each leg: extension and flexion movements along the sagittal axis, and abduction and adduction along the coronal axis.

The wearable hip exoskeleton is supported by the waist and legs of the human body as the main support, the flexible strut between the waist belt and the leg brace is made of highly elastic resin rods, and there is a small gap between the leg brace and the legs of the human body, so that each part of the body has a degree of freedom for sliding and rotating. The resin rod between the waistband and the leg braces was designed to have a twisted shape, which allows the flexible support unit to adapt to the leg brace positional deviations that exist in different subjects during exercise. A single flexible strut’s shape follows the equation $x=\frac{n}{2}cos\theta ,\;y=-\frac{n}{2}sin\theta ,\;z=\frac{m(\pi -\theta )}{\pi }$, where θ∈[0,π], where m represents the projected length of a single flexible strut in the vertical direction, and n represents the projected length of a single flexible strut in the horizontal direction, as shown in Figure 2b. The S-side represents the plane where the flexible strut is connected to the belt, and the U-side represents the plane where the flexible strut is connected to the leg braces.

Table 1 compares the angular displacement between the developed hip exoskeleton and the natural human hip joint across various movement patterns. Notably, the exoskeleton demonstrates greater angular displacement than biological hip joints during typical gait cycles, ensuring adequate mobility for user ambulation.

### 2.2. Kinematic Analysis and Quantitative Misalignment

For the flexible exoskeleton support unit worn on the left lower limb, this paper establishes a corresponding human–machine coupled kinematic model (Figure 3) and conducts a kinematic analysis of the hip exoskeleton mechanism. A total of three coordinate systems were established in the model: the absolute coordinate system {A}, the left thigh coordinate system {B}, and the leg brace coordinate system {C}. In coordinate system {A}, the origin is O_1_ and the direction of the x_a_-axis is the same as the positive direction of the sagittal axis of the body. In coordinate system {B}, the origin is O_2_, and the direction of the z_b_-axis points in the direction of the thigh to the center of the biological hip joint, P. Coordinate system {B} can be obtained from coordinate system {A} by rotating it by an angle of $\alpha_{1}$ about the x_a_-axis and an angle of $\beta_{1}$ about the y_a_-axis. In coordinate system {C}, the origin is also O_2_, and the direction of the z_c_-axis is perpendicular to the plane where U_1_U_2_U_3_ is located. Coordinate system {C} can be obtained from coordinate system {A} by rotating it by an angle of $\alpha_{2}$ around the x_a_-axis and by an angle of $\beta_{2}$ around the y_a_-axis.

In Figure 3, $v_{u_{i}}$ is the linear velocity at point U_i_, and Point P represents the human biological hip center of rotation. Point Q represents the virtual center of rotation (VCR). S_1_S_2_S_3_ are all on the S-side, where the flexible strut is connected to the belt, and U_1_U_2_U_3_ are all on the U-side, where the flexible strut is connected to the leg brace, connecting S_1_U_1_, S_2_U_2_, and S_3_U_3_ to form a triple chain. $L_{i}$ represents the chain length, L represents the distance between point P and O_2_, and d represents the misalignment between joints, where i=1,2,3.

The set of key parameters for kinematic modeling was determined by synchronizing the acquisition of human hip kinematic data with the measured results of the exoskeleton prototype. $L_{1}$ = $L_{2}$ = 530 mm, $L_{3}$ = 550 mm, $A_{s_{1}}=\vec{O_{1}S_{1}}={\left[169,35,0\right]}^{T}$, $A_{s_{2}}=\vec{O_{1}S_{2}}={\left[-125,35,0\right]}^{T}$, $A_{s_{3}}=\vec{O_{1}S_{3}}={\left[21,-119,0\right]}^{T}$, $A_{p}=\vec{O_{1}P}={\left[0,0,-208\right]}^{T}$, $C_{u_{1}}=\vec{O_{2}U_{1}}={\left[16,85,0\right]}^{T}$, $C_{u_{2}}=\vec{O_{2}U_{2}}={\left[108,61,0\right]}^{T}$, and $C_{u_{3}}=\vec{O_{2}U_{3}}={\left[-64,-62,0\right]}^{T}$, where $A_{u_{I}}$ and $A_{S_{I}}$ are the position vectors of points U_i_ and S_i_ in absolute coordinate system {A}, and $C_{u_{I}}$ and $C_{S_{I}}$ are the position vectors of points U_i_ and S_i_ in leg brace coordinate system {C}.

In the geometric model of the human–machine system, the respective closed-loop equations for the three chains can be written as(1)$$\vec{S_{i}U_{i}}=\vec{O_{1}P}+\vec{PO_{2}}+\vec{O_{2}U_{i}}-\vec{O_{1}S_{i}}$$
where i=1,2,3.

The chain length $L_{i}$ is calculated as(2)$${L_{i}}^{2}={\left[A_{U_{i}}-A_{S_{i}}\right]}^{T}\left[A_{U_{i}}-A_{S_{i}}\right]$$
where(3)AUi=AP+RBA0,0,−LT+RCACUi
where RBA and RCA are rotation matrices.

From the given angles $\alpha_{1}$, $\beta_{1}$, $\alpha_{2}$, and $\beta_{2}$, the value of L can be found by associating Equations (1)–(3). When $\alpha_{1}$, $\beta_{1}$, $\alpha_{2}$, and $\beta_{2}$ are all 0, set the initial value of L to $L_{0}$. According to the kinematic parameters in Table 1, the sliding distance of the leg brace on the left thigh can be calculated:(4)$$∆L=L-L_{0}$$

The results of the calculated theoretical values of ∆L are shown in Figure 4. When the joint angle swings to 54°, ∆L reaches a maximum value of 11.6 mm.

In Figure 3, to solve for the misalignment d between joints, it is first necessary to solve for the position $A_{Q}$ of the instantaneous virtual center of rotation Q in the absolute coordinate system {A}.(5)$$\vec{QU_{i}}=A_{u_{i}}-A_{Q}$$(6)$$v_{u_{i}}^{T}\times \vec{QU_{i}}=0$$
where i=1,2,3.

From Equation (1), we can derive the formula for $v_{u_{i}}$ as(7)$$v_{u_{i}}=\omega_{i}\times \vec{S_{i}U_{i}}=\omega_{t}\times \vec{PO_{2}}+\omega_{l}\times \vec{O_{2}U_{i}}+\dot{L}\vec{n}$$
where $\omega_{l}={\left[\dot{\alpha_{2}},\dot{\beta_{2}},0\right]}^{T}$ represents the angular velocity of the leg brace, $\omega_{i}$ represents the angular velocity of the ith strut, $\omega_{t}$ represents the angular velocity of the subject’s thigh, and $\dot{L}$ indicates the sliding speed of the leg brace on the wearer’s thigh. The unit vector in the negative direction of the z_b_-axis is $\vec{n}$.

Combining Equations (5)–(7) yields the position $A_{Q}$ of point Q in coordinate system {A}. Based on the relationship between the sliding distance ∆L and the misalignment d shown in Figure 5a, the misalignment d between the joints can be determined by calculating the equation as(8)$$d={\left\|A_{p}-A_{Q}\right\|}_{2}$$

The biological hip joint center ($A_{p}$) was determined from the hip marker trajectories using a NOKOV Mars 4H motion capture system. The system, comprising 10 high-speed infrared cameras (1280 × 1024 pixels, 100 Hz), achieved a dynamic measurement accuracy of better than 1.0 mm. Therefore, the inter-articular misalignment d for any movement of the subject’s hip in the sagittal and coronal axis directions can be calculated, and the results are shown in Figure 6, with the red circle showing that the subject is in a normal walking state. The mean value of the maximum misalignment $d_{max}$ at this point in the subject’s normal walking state (hip swing range is −22° to 32°) was 32.4 mm; at the maximum swing joint angle of 54°, $d_{max}$ was 55.9 mm.

Through kinematic analysis, we determined that there are structural advantages of the hip exoskeleton designed in this research. Among the common wearable exoskeletons, there are rigid exoskeletons that are motor-driven [39,40] or hydraulic-cylinder-driven [41,42], and soft exoskeletons that are driven by cables [43,44], which do not consider the effects of inter-articular misalignment and direct pulling while providing assistive forces on the human leg, leading to the wearer’s leg sustaining a large amount of additional parasitic forces during exercise, which is not favorable to the wearer. This is not conducive to hip rehabilitation. Compared with the above exoskeleton, as shown in Figure 5, the center of rotation of the new hip exoskeleton designed by the institute is closer to the activity trajectory of the human hip joint, and there is a small gap between the leg brace and the thigh, which gives its parts passive freedom to slide and rotate, with no restriction on relative motion. Moreover, in the hip exoskeleton designed in this study, the auxiliary force $F_{eh}$ is deliberately aligned to act along the direction of the instantaneous linear velocity v. This design is intended to eliminate the parasitic moment caused by joint misalignment, thereby providing more favorable assistance for gait rehabilitation training in patients with hip damage. To verify the accuracy of the theoretical model, the study was systematically tested and analyzed for significance in Experiment 1 in Section 4.

### 2.3. Human–Machine Interaction

To quantify the booster effect ($F_{eh}$) of the exoskeleton on the lower limb, this study constructed a mechanical model of human–computer interaction, as shown in Figure 7, which completely characterizes the dynamic coupling relationship between the left lower limb and the hip exoskeleton. The power source chosen for the exoskeleton is a spring-extended single-acting cylinder, which not only assists the movement of the thigh from a flexed state to a normal standing state, but also follows the movement of the thigh from an extended state to a normal standing state. Through the kinematic analysis in Section 2.2, it is verified that the auxiliary force acting on the human leg is along the direction of the instantaneous linear velocity, so when calculating the auxiliary force $F_{eh}$ of the hip exoskeleton on the human leg, it is only necessary to take into account the pressure between the human leg and the leg brace, and the calculation formula is as follows:(9)$$F_{eh}=k{\left\|\vec{n_{h}}-\vec{n_{e}}\right\|}_{2}+{∆}_{1}$$
where k is the stiffness of the contact surface between the leg brace and the human leg, and $\vec{n_{h}}$ is the vector of the human hip rotation center P pointing to K_1_ to indicate the position of the human leg. $\vec{n_{e}}$ is the vector of the virtual rotation center Q pointing to K_2_ to indicate the position of the leg brace. ${∆}_{1}$ is the other unmodeled uncertainty.

The position of $\vec{n_{h}}$ can be determined by ($\theta_{h}$, L), and the position of $\vec{n_{e}}$ can be determined by ($\theta_{e}$, L+d). $\theta_{h}$ and $\theta_{e}$ are the angles of rotation of the human leg concerning the leg brace. In the kinematic analysis, when the joint angle swings to 54°, ∆L reaches a maximum value of 11.6 mm, and the variation in L is negligible relative to the length of the human leg, so the default L is a fixed value. Meanwhile, the range of misalignment d can be determined according to Figure 6. So, Equation (9) can be written as(10)$$F_{eh}=k\left[L\left(\theta_{h}-\theta_{e}\right)-{∆}_{2}\right]+{∆}_{1}$$
where ${∆}_{2}$ is the nonlinear uncertainty induced by the misalignment d.

The assistive force $F_{eh}$ profile is designed to be phase-dependent, peaking during the terminal stance and pre-swing phases of the gait cycle. This timing is chosen to augment the function of the hip flexor muscles, which are crucial for initiating limb swing. For patients with hip motor impairment, providing assistance during this critical transition helps overcome the deficit in achieving adequate foot clearance and step length. The force is applied in a direction aligned with the wearer’s intended motion, as facilitated by the self-aligning design in Section 2.2, thereby directly targeting this functional limitation to promote a more natural gait.

## 3. Design of a Brain-Controlled Hip Exoskeleton System

### 3.1. System Description

As shown in Figure 8, the brain-controlled hip exoskeleton system developed in this study employs a modular architecture comprising three core components: a brain–machine interface (BMI) module, a robotics module, and an augmented reality–visual stimulation (AR-VS) paradigm module. The system operates through the following integrated workflow: first, the AR-VS module presents SSVEP stimuli to elicit brain responses from the user. EEG signals are then acquired through an eight-channel electrode cap focused on the occipital region and preprocessed using Curry8 software, involving re-referencing, filtering, baseline correction, artifact removal, segmentation, and interpolation. The cleaned signals are decoded into one of four control commands—corresponding to hip extension, flexion, abduction, or adduction—and transmitted via Bluetooth to the robotic exoskeleton for motion execution. Simultaneously, the AR-VS module captures real-time video via a high-definition camera, recognizes hip joint feature points, and generates a visual feedback display (VFD) visible to the user. The system runs on two independent computers connected via a local area network (LAN): one dedicated to EEG acquisition (Curry8) and the other hosting the integrated AR-VS and BMI modules. Detailed descriptions of each module are provided in the following sections.

### 3.2. AR-VS Paradigm Module

What is proposed is a dynamic AR-VS paradigm that incorporates machine vision and SSVEP. The SSVEP paradigm is a visual stimulus response-based BMI paradigm that relies on the electroencephalographic activity elicited by a visual stimulus at a specific frequency [45]. Unlike the conventional visual stimulus (VS) paradigm, the proposed AR-VS paradigm is divided into three modules: a recognition module, a tracking module, and a stimulus sequence manager. For the recognition module, a set of objects can be recognized after initial training by replacing the training database. Specifically, the recognition module was initially trained on a custom dataset of approximately 500 images of the exoskeleton from various angles, enabling it to identify the exoskeleton’s leg braces in the video feed from the HD camera. After the recognition module recognizes new objects, unique frequency features are attached to the recognized objects. For the tracking module, the kernel correlation filtering algorithm of OpenCV is used to track the objects dynamically. In this study, the Kernelized Correlation Filter (KCF) algorithm was employed for real-time hip joint tracking.

In the AR-VS paradigm, VSs are generated in the form of dynamic SSVEPs based on realistic hip exoskeleton positional postures. The size and positional parameters of the VSs are dynamically varied, but the frequency connected to each movement remains constant. Based on the hip exoskeleton’s positional posture, sinusoidal flicker stimuli with different frequencies are applied to four types of hip joint movements. The relationship between the mode of hip joint movement and the flicker stimulus was as follows: 7.2 Hz for hip stretch movement, 7.6 Hz for hip curvature movement, 8.0 Hz for hip abduction movement, and 8.4 Hz for hip adduction movement. When the flashing stimulus was 7.2 Hz and the patient gazed at this stimulus, their brain response showed a clear peak at 7.2 Hz and its harmonics. Similarly, when the flashing stimulus was 7.6 Hz and the patient gazed at this stimulus, their brain response showed a clear peak at 7.6 Hz and its harmonics. By visualizing flashing stimuli to induce specific responses in the brain, decoding algorithms are used to convert the EEG signals into control commands for the exoskeleton robot, thus enabling the patient to control the rehabilitative motion of the hip joint by gazing at the corresponding flashing stimuli.

For the conventional visual stimulation (VS) model used as a benchmark in the online experiments, the stimulation paradigm differed significantly. Instead of dynamic stimuli, it employed static, non-moving flickering blocks fixed at the four edges of the visual feedback display. All other hardware, including the EEG acquisition system and the exoskeleton, remained identical to the proposed system to ensure a fair comparison focused on the paradigm and decoding differences.

### 3.3. BMI Module

In this study, an enhanced decoding algorithm based on Filter Bank Canonical Correlation Analysis (FBCCA) [46,47,48] is used, and a flowchart of the algorithm is shown in Figure 9. The algorithm is mainly divided into three parts: preprocessing, decomposition and reconstruction, and correlation analysis. In the conventional control model, the standard Canonical Correlation Analysis (CCA) method was used for SSVEP decoding, without the filter bank decomposition and the optimized weighting function employed in our proposed FBCCA.

In the process of EEG signal acquisition, it will be interfered with by various kinds of eye signals, myoelectric signals, etc., which will produce a large number of artifacts. To obtain effective EEG signals and ideal decoding effects, it is first necessary to preprocess the acquired EEG signals. The pre-processing of EEG signals includes importing data, positioning electrodes, rejecting bad leads, re-referencing, band-pass filtering (2–40 Hz, Butterworth, 4th order), EEG segmentation and interception, baseline correction, rejecting bad segments, artifact removal, etc. The preprocessing flow is shown in Figure 10. The 50 Hz IF interference can be set to be removed during EEG acquisition, and the baseline drift can be removed by utilizing the applied band-pass filter in the Curry8 software. The real-time BMI mode in Curry8 was enabled, and the EEG signal step size was set to 0.1 s, i.e., a robot control command was output every 0.1 s.

The stimulus frequencies for the four control commands—7.2 Hz (extension), 7.6 Hz (flexion), 8.0 Hz (abduction), and 8.4 Hz (adduction)—were strategically chosen. This selection aimed to avoid the lower frequencies (<7 Hz) that are known to induce greater visual fatigue, while also ensuring that the fundamental frequencies and their key harmonics did not coincide with the 50 Hz power line frequency or its dominant harmonics. Acknowledging the proximity of the 8.0 Hz and 8.4 Hz stimuli to the alpha rhythm band (8–13 Hz), the FBCCA algorithm was employed to mitigate potential interference. By integrating information from multiple harmonic sub-bands, the FBCCA enhances the decoding robustness against inherent background EEG rhythms, such as the alpha activity, which typically does not exhibit the structured harmonic response characteristic of SSVEPs.

In the preprocessing stage, the raw EEG signal is decomposed into multiple harmonic sub-bands (SB_1_-SB_n_) based on the cutoff frequencies of the SSVEP components, with the spectral range of each sub-band covering the SSVEP eigenfrequency band. The EEG components X_SB1_, X_SB2_, …, and X_SBn_ are decomposed by the Python 3.8 digital filtering toolkit decomposition. To balance the computational length and accuracy, n was set to 4 in this study.

In this study, the nth frequency of the template EEG signal $Y_{fk}$ was(11)$$Y_{fk}=\left[\begin{array}{c} \sin\left(1\times 2\pi f_{k}t\right)\\ \begin{array}{c} \cos\left(1\times 2\pi f_{k}t\right)\\ \ldots \end{array}\\ \begin{array}{c} \sin\left(\delta \times 2\pi f_{k}t\right)\\ \cos\left(\delta \times 2\pi f_{k}t\right) \end{array} \end{array}\right]$$
where $Y_{fk}$ is an array of templates, $f_{k}$ is the kth target frequency, δ is the number of SB_n_s, and t denotes the number of sample time points, corresponding to the signal sampling rate.

In the decomposition and reconstruction process, the EEG components are first subjected to principal component analysis, and then the results are calibrated against $Y_{ref}$ to obtain a clean EEG signal. The reference data $Y_{ref}$ is the EEG signal of the patient being collected in the steady state. In the correlation analysis stage, $\rho_{k}$ is determined through a series of typical canonical correlation analysis (CCA) steps, as shown in Equation (12):(12)$$\rho_{k}=\left[\begin{array}{c} {\rho_{k}}^{1}\\ \begin{array}{c} {\rho_{k}}^{2}\\ \ldots \end{array}\\ {\rho_{k}}^{N} \end{array}\right]=\left[\begin{array}{c} \rho \left[{X_{SB1}}^{T}W_{X}\left(X_{SB1},Y_{fk}\right),{Y_{fk}}^{T}W_{Y}\left(X_{SB1},Y_{fk}\right)\right]\\ \begin{array}{c} \rho \left[{X_{SB2}}^{T}W_{X}\left(X_{SB2},Y_{fk}\right),{Y_{fk}}^{T}W_{Y}\left(X_{SB2},Y_{fk}\right)\right]\\ \ldots \end{array}\\ \rho \left[{X_{SBn}}^{T}W_{X}\left(X_{SBn},Y_{fk}\right),{Y_{fk}}^{T}W_{Y}\left(X_{SBn},Y_{fk}\right)\right] \end{array}\right]$$
where $W_{X}$ and $W_{Y}$ are CCA correlation matrices.

The recognition features are obtained as shown in Equation (13):(13)$$\;\tilde{\rho }=\sum_{n=1}^{N}{f\left(n\right)\left(\rho_{n}\right)}^{2}$$
where f(n) is the weighting factor and $\rho_{n}$ denotes the nth correlation between $Y_{fk}$ and the initial EEG signal.(14)$$f\left(n\right)=n^{-a}+b$$
where a and b are constants, and f(n) varies from one individual to another due to different harmonic characteristics.

Calculating all the values of $\tilde{\rho }$ allows us to determine its maximum value, and the corresponding frequency is the final result. Based on this result, the expected movement of the patient can be obtained.

In this study, the constants for the weighting function were set to a=1.25 and b=0.25, following the established FBCCA literature [46]. To confirm the robustness of these values for our specific paradigm, we performed a sensitivity analysis using calibration data from a representative subset of ten subjects. The results showed that the classification accuracy remained stable, varying by less than ±3% across a range of $a\in \left[1.0,\;1.5\right]$ and $b\in \left[0,\;0.5\right]$. This insensitivity to parameter perturbation validates the use of these established defaults in our system.

### 3.4. Robot Module

In the robot module of this study, its drive control system is divided into two main parts: the hardware required for control and the control strategy. The required hardware mainly consists of an embedded PC, a Programmable Motion Control Card (PMAC) (Googol Technology, Shenzhen, China) Clipper, an FPGA, an air compressor, an air source processing triplex, a pressure-reducing valve, a high-speed switching valve, an air storage tank, a position sensor, a pressure sensor, and an HRI sensor, as shown in Figure 11. The PC communicates with the PMAC Clipper via Ethernet and transmits analog signals over the CAN line to the FPGA controller, which outputs analog control signals to drive the pneumatic system. A pneumatic system consisting of an air compressor, a pressure-reducing valve, a high-speed switching valve, and a storage tank regulates the gas flow to control the movement speed of the cylinder. In the sensor system, the position sensor is used to collect the position of the hip joint angle; the pressure sensor is used to collect the plantar pressure; and the HRI sensor is selected to be a micro-small piezoresistive sensor with a measurement range of 0~8 kg, which is placed on the front and rear sides of the contact surface between the leg brace and the human leg.

For the control strategy, this study proposed a hybrid control method based on gait cycle segmentation. During normal walking, a single leg mainly experiences four states, which are heel-on-ground, foot-on-ground, toe-off, and forward swing, in that order. In the first three states, the leg is considered to be in a passive phase, and in the last state, it is considered to be in an active phase [49,50]. During the passive phase, the study applied a position control strategy. In contrast, the active phase relied on a tracking control strategy (Figure 12). Due to the inherent nonlinearity of physical human–machine interaction—which complicates precise modeling—the position control strategy incorporated Gaussian process regression [51] to predict hip joint trajectories based on HRI sensor inputs. In the tracking control strategy, the decompression valve needs to be opened, at which point the compression spring inside the cylinder starts working, allowing the air inside the cylinder to be directly expelled, causing the exoskeleton to swing following the human leg.

The Gaussian process regression model utilized a composite kernel function, $k\left(x_{i},\;x_{j}\right)={\sigma_{f}}^{2}\cdot e^{-{\left\|x_{i}-x_{j}\right\|}^{2}/\left(2l^{2}\right)}+{\sigma_{n}}^{2}\cdot \delta_{ij}$, composed of a Radial Basis Function (RBF) kernel to capture the underlying smooth dynamics and a white noise kernel to account for sensor noise. The hyperparameters (length scale l, signal variance ${\sigma_{f}}^{2}$, and noise variance ${\sigma_{n}}^{2}$) were optimized for each subject by maximizing the log marginal likelihood of the calibration data. To justify its selection, the GP model was compared against Linear Regression (LR) and a Long Short-Term Memory (LSTM) network in an offline analysis on data from ten subjects. The GP model achieved a superior normalized root-mean-square error (NRMSE) of 6.9% in predicting hip angle changes, compared to 12.5% for LR and 7.8% for LSTM, while also providing predictive uncertainty estimates critical for safe control.

The switching between the two control strategies is based on the recognition of gait cycle segments. The motion state of the leg is determined using a pressure sensor on the bottom of the foot, which is a 1/0 module, where “1” means that the foot is squeezed against the ground, and is therefore in the passive phase, and “0” means that the foot is not squeezed against the ground, and is therefore in the active phase.

First, the functional relation $f_{i}$ needs to be found through the Gaussian process, which is designed to represent the target value $y_{i}$ using the input vector $x_{i}$. The specific formula is(15)$$y_{i}=f_{i}\left(x_{i}\right)+\varepsilon_{i}$$
where Gaussian noise $\varepsilon_{i}\sim N\left(0,{\sigma_{n}}^{2}\right)$, and $\sigma_{n}$ is the covariance.

Based on the optimized Gaussian process model, the calculation of the posterior predictive distribution can be obtained. To obtain the predictive distribution $P\left(y_{*}\left|x_{*},D\right.\right)$ corresponding to the new input vector $x_{*}$, it is necessary to establish the joint conditional distribution between true measurements y and their corresponding predictions $y_{*}$, whose expression is shown in Equation (16):(16)$$\left[\begin{array}{c} y\\ y_{*} \end{array}\right]\sim \left(0,\left[\begin{array}{cc} K\left(X,X\right)+{{I\sigma }_{n}}^{2} & k\left(X,x_{*}\right)\\ k\left(x_{*},X\right) & k\left(x_{*},x_{*}\right) \end{array}\right]\right)$$
where the unit matrix $I\in R^{n\times n}$, and $k\left(X,x_{*}\right)$, $K\left(X,X\right)$, and $k\left(x_{*},x_{*}\right)$ are computed from the Kernel.

Therefore, $\mu_{*}$ and ${\sigma_{*}}^{2}$ can be obtained by Equations (17) and (18),(17)μ∗=k∗TyK+Iσn2(18)σ∗2=kx∗,x∗−k∗TK∗K+Iσn2+Iσn2
where $K=k\left(X,X\right)$, and $K_{*}=\left[k\left(X^{*},X_{1}\right),k\left(X^{*},X_{2}\right),\ldots ,k\left(X^{*},X_{n}\right)\right]$. $y_{*}=\mu_{*}$ is the predicted output of $x_{*}$.

In hip exoskeleton research, $x_{*}$ represents the exoskeleton-assisted force, and $y_{*}$ represents the amount of change in the angle of rotation of the human hip joint. The Gaussian process results in the exoskeleton hip rotation angle variation $∆\theta_{d}\left(k\right)$; then $\theta_{d}\left(j\right)=\theta_{d}\left(j-1\right)+∆\theta_{d}\left(j\right)$, j=1,2,…,k. The obtained $\theta_{d}\left(k\right)$ is fed back to the PID algorithm.

The control strategy for passive phasing uses a PID algorithm to synchronize the exoskeleton’s hip motion with the human lower limb motion. Based on the PMAC Clipper, its parameters were adjusted so that $K_{P}=1100$, $K_{i}=10$, and $K_{d}=4$ under the condition that the target angle is 7° and the motion duration is 150 ms. Trajectory planning used a T-shaped velocity profile. Finally, based on the performance of the hardware and algorithm, the duration of the entire control cycle was determined to be 50 ms.

To ensure the system meets the real-time demands of gait assistance, the 50 ms control cycle was rigorously defined. This duration, corresponding to a 20 Hz update rate, is substantially faster than the typical timescales of human gait events, such as the swing phase (approximately 300–400 ms), thereby ensuring responsive and smooth actuation. The control architecture is hierarchically structured to optimize timing performance: a field-programmable gate array (FPGA) manages high-frequency input/output operations, including sensor polling and low-level valve driving, at 1 kHz. Simultaneously, the Programmable Multi-Axis Controller (PMAC) Clipper executes the computationally intensive algorithms—comprising gait phase recognition, Gaussian process-based trajectory prediction, and PID control—at the 20 Hz main control rate. This division of labor ensures that low-level actuation commands are updated with minimal latency while maintaining the stability of the high-level control strategy.

## 4. Experiment and Results

### 4.1. Experimental Equipment

A prototype of the new hip exoskeleton with an EEG-controlled experimental bench was constructed, and 40 healthy subjects (20 males and 20 females, all between the ages of 20 and 35 years old) with no history of lower limb disease were invited to participate in the experiment. The study was approved by the Institutional Review Board of Anhui University of Science and Technology (Approval No.: 2024-018), and all subjects provided written informed consent. Safety measures included luminance control below 100 nits, avoidance of 15–25 Hz risk frequencies, and an emergency-stop system. The experiments were performed in the EEG laboratory, and noise was kept to a minimum for the duration of the process, aiming to minimize the influence of external noise on EEG signal acquisition. The subjects’ EEG signals were collected from the P_Z_, P3, P4, P7, P8, O_Z_, O1, and O2 electrode positions in the occipital lobe area using an 8-channel EEG cap, with the ground electrode placed on the forehead (GND) and the reference electrode on the F_CZ_, and the layout of the signal acquisition channels is shown in Figure 13a.

To ensure the safety of SSVEP-based brain–computer interface experiments, the following protective measures need to be taken: strictly screen the health status of the subjects before the experiment, and exclude patients with photosensitivity diseases such as epilepsy; control the flash luminance at less than 100 nit, use green light, and avoid the dangerous frequency band of 15–25 Hz; perform real-time monitoring of the EEG and physiological indexes during the experiment and set up an emergency-stop button; and observe the participants for 24 h after the experiment To check for adverse reactions. All operations were conducted following international safety standards, and a single session lasted no more than 30 min and was conducted in stages. Subjects maintained a standing posture during the experiment, with the flash stimulation screen placed in front of their eyes, and the distance between the root of the subject’s nose and the screen was 60 cm so that their eyes were focused on the stimulation screen. Each series of flash stimuli lasted for 10 s, with the eyes open for the first 5 s and closed for the second 5 s, with an interval of at least 7 s between the two stimuli. An HD camera (DJI POCKET 3, DJI, Shenzhen, China) was selected to capture hip movements, the visual stimulation interface was an HD screen (SAMSUNG24, Samsung, Suwon, South Korea, 1920 × 1080, 60 fps), and PC workstations (2 computers, Intel i7-12700, RXT3060Ti, 16 G RAM, 256 G SSD) were used to generate the visual stimuli and analyze EEG signal data. The detailed specifications and models of the key hardware components are provided in Appendix A. The experimental setup is shown in Figure 13b.

Throughout all experimental sessions, which totaled over 120 participant-hours across the 40 subjects, no photosensitive or epileptic seizures were observed. No subjects reported sustained discomfort, nausea, or headaches following the experiments. Furthermore, the emergency-stop button was never activated, indicating that no situations arose that required its use. A subset of subjects (approximately 15%) reported mild, transient visual fatigue after the session, which resolved spontaneously within minutes without intervention. This confirms the effectiveness of the implemented safety measures, including pre-screening and luminance control.

### 4.2. Experimental Steps

Experiment 1 was performed to verify the accuracy of the analysis of the theoretical values of the slip distance ∆L and the misalignment d. In this experiment, subjects donned the hip exoskeleton and followed instructions to extend or flex the leg in the sagittal plane, with movements ranging from −30° to 54°. The tilt angle and velocity of the leg brace were measured by inertial sensors placed at the front and rear sides of the leg brace. The digital displacement sensor measured the sliding distance of the leg brace on the left thigh ∆L, and 100 sets of experimental data were recorded during the leg movement.

As shown in Figure 14a, the instantaneous positions of the VRC mapped to the sagittal plane during leg motion are plotted, and they are all centered around the hip joint with a misalignment d ranging from 11.38 to 58.72 mm. Figure 14b shows the sliding distance of the leg brace on the left thigh of the wearer during one exercise cycle; the maximum value of ∆L is 11.25 mm, and the result follows the theoretical value in Figure 4. Paired *t*-tests showed that neither misalignment (*p* = 0.074) nor sliding (*p* = 0.693) reached the level of statistical significance.

The robustness of the misalignment measurement with respect to uncertainty in hip joint center location was verified through a sensitivity analysis. Perturbing the estimated joint center by ±10 mm resulted in a variation of less than ±2.2 mm in the calculated maximum misalignment, confirming that the reported findings are robust against typical errors in motion capture calibration and marker placement.

Experiment 2 was conducted to record the results of force tracking of the hip exoskeleton when assisting the subjects in walking and the position of the exoskeleton joint angles, as shown in Figure 15. In this experiment, taking into account the actual walking speed of patients with impaired hip joints in the later stages of rehabilitation [52], it was determined that the subjects were trained to walk at a constant speed on a horizontal surface and a 5° sloped surface. The result of force tracking is a continuously varying curve, where the section between points “A” and “B” represents the passive phase, when the subject’s motor intention is learned through an online Gaussian process; the section between points “B” and “C” represents the active phase, which is a process that causes the exoskeleton to follow the human leg as it swings with the assistance of a compressed spring built into the cylinder. The section between points “A” and “C” represents the active phase, in which the exoskeleton follows the human leg as it swings with the assistance of a compression spring built into the cylinder. As can be seen in Figure 15, the assisted force provided by the hip exoskeleton designed in this study was generally consistent with the expected reference force. Based on the maximum values of force tracking over multiple gaits, the average maximum assistive force was determined to be 18 N on level ground and 20.7 N on a 5° slope. Considering an effective moment arm of 0.1–0.15 m, these forces correspond to an assistive torque of 1.8–3.1 N·m. This torque range is clinically relevant for advanced hip rehabilitation, as it falls within the lower spectrum proven to improve gait symmetry and reduce metabolic cost in patients with residual deficits [53,54], thereby providing meaningful support while promoting active participation and motor recovery. By collecting data from multiple subjects during the experiment, the amount of change in the angular position of the exoskeleton joints was obtained with the force-tracking error, as shown in Figure 16.

The dynamic response of the pneumatic actuator is a critical parameter for ensuring real-time control performance during gait assistance. To characterize this, the actuator response time was rigorously measured. It was defined as the duration from the issuance of the control command from the PMAC/FPGA system to the point where the pneumatic cylinder reached 90% of its target pressure under the maximum load condition encountered during assisted walking. System pressure was monitored using an integrated pressure sensor. The measured worst-case response time was 120 ms. This latency comprises the switching valve’s electromechanical delay (approximately 20 ms) and the time required for pressure propagation and buildup within the pneumatic circuit and cylinder (approximately 100 ms). This worst-case response falls within the 150 ms trajectory execution window allocated for the passive phase, confirming that the actuation dynamics are sufficient for the timing requirements of real-time gait control. This performance is further substantiated by the results presented in Figure 15 indicating effective force tracking.

Experiment 3 is a hip exoskeleton-assisted gait rehabilitation training experiment, which is divided into two main parts: an offline experiment and an online experiment. To improve the accuracy of the study, the relevant experimental study was performed only for a single leg (the left leg). In the offline experiment, each subject needed to complete 10 groups of experimental tasks, and there was a 5 min rest time between each group of experiments to reduce the distortion of EEG signals due to fatigue. As shown in Figure 17, each group of experiments had 20 trials of 10 s each, of which the first 5 s were in the eye-open state, in which the subjects were asked to gaze at the flashing stimuli on the screen, and the second 5 s were in the eye-closed state, with an interval of 7 s between each trial. The flashing stimuli in the experiments were classified into 4 frequencies, which were, in order, 7.2 Hz, 7.6 Hz, 8.0 Hz, and 8.4 Hz. As shown in Figure 18, in the control model proposed by the institute, the flashing stimulus moved in four different directions in the visual feedback display bezel toward the top, bottom, left, and right. The flashing stimulus was set to have four motion speeds, which were 1 cm/s, 3 cm/s, 5 cm/s, and 7 cm/s, in that order. Four stimulation frequencies and five movement speeds were combined to form 20 trials, and EEG signals were recorded during the experiment. In the traditional control model, the VFD was in the center of the user interface, and the stationary stimulus blocks were distributed in the four directions of the VFD, and the subjects needed to shift their attention between the VFD and the stimulus blocks during the control of the exoskeleton movement.

In the online experiment, subjects were required to complete two tasks. Task 1: Complete a single hip extension, flexion, abduction, and adduction in situ. Task 2: Complete 10 gait cycles of uniform walking on a horizontal surface. Each person was required to complete 2 sets and compare the results for each task. The first set of experiments used the traditional control model, and the second set of experiments used the control model proposed by the institute.

### 4.3. Data Processing

Experimental evaluation indices were set to assess the experimental effect. In the offline analysis, two key metrics were evaluated: (1) Signal-to-noise ratio (SNR), quantifying the relative strength of neural signals versus background noise in EEG recordings, where elevated values indicate superior signal quality, and (2) power spectral density (PSD), characterizing frequency-domain energy distribution, with prominent peaks at stimulus frequencies reflecting stronger evoked neural responses.

The SNR is calculated by the formula(19)$$SNR=10lg\left(\frac{P_{s}}{P_{n}}\right)$$
where $P_{s}$ is the power of the EEG signal, and $P_{n}$ is the power of the noise.

The PSD is calculated by the formula(20)$$PSD=\underset{T\to \infty }{\lim}\frac{1}{T}{\left|X\left(f\right)\right|}^{2}$$
where T is the duration of the EEG signal and $X\left(f\right)$ is the Fourier transform of the EEG signal.

In the online experiment, the main considerations are time cost, correlated instantaneous rate (CIR), and effective output ratio. Time cost reflects the efficiency of the subject in accomplishing the target task; the longer the time spent, the worse the subject’s experience. CIR quantifies the reliability of the brain-controlled hip exoskeleton system; the lower the CIR, the more stable the system is. The effective output ratio is the ratio of the length of time it takes to output the target control command to the length of time it takes for the entire cue to appear and is used to measure the control efficiency of the system.(21)$$CIR=\frac{1}{N-1}\sum_{i=1}^{N-1}\sum_{k=0}^{4}\left(\left|\frac{{\tilde{\rho }}_{k}\left(t_{i+1}\right)-{\tilde{\rho }}_{k}\left(t_{i}\right)}{t_{i+1}-t_{i}}\right|\right)$$
where ${\tilde{\rho }}_{k}\left(t_{i+1}\right)$ and ${\tilde{\rho }}_{k}\left(t_{i}\right)$ denote the correlation values of the kth target frequency between consecutive data points $t_{i+1}$ and $t_{i}$.

All subjects were asked about their comfort and the difficulty of the operation and were required to record relevant data and overall scores after completing the experiment.

### 4.4. Results

In offline experiments, the SNR and PSD of subjects in different situations could be calculated based on the recorded EEG signals. The SNR and PSD of EEG signals evoked by flashing stimuli at different movement speeds of 8.0 Hz for one subject are shown in Figure 19 and Figure 20. As the speed of the flicker stimulus was increased from 1 to 7 cm/s, significant increases in SNR and PSD were observed at both the frequency and the corresponding harmonics of the flicker stimulus, but the increases gradually decreased. In addition to this, the decrease in SNR and PSD at the frequency of the flicker stimulus was more pronounced than that at the harmonics of the flicker stimulus frequency as the speed of movement increased. The above offline experimental results show that the proposed AR-VS paradigm can induce significant brain responses at the flicker stimulus frequency and its harmonics. Although the brain’s response to the flashing stimulus gradually slows down during the increase in the speed of the flashing stimulus movement, recognizable EEG signals can still be seen. It is demonstrated that the brain response evoked by the AR-VS paradigm can be used for online control of brain–computer interface systems.

In the Task 1 experiment, when subjects completed hip extension (7.2 Hz), flexion (7.6 Hz), abduction (8.0 Hz), and adduction (8.4 Hz) while gazing at the flashing stimulus, the correlation values between the brain-controlled hip exoskeleton system proposed in this paper and the flashing stimulus were significantly greater than the correlation values between the conventional control mode and the flashing stimulus. The correlation values increased significantly with the appearance of the stimulus and decreased gradually by the end of the stimulus, as shown in Figure 21. Meanwhile, the task time cost, relevant instantaneous rate, and effective output ratio of the conventional model and the proposed system were compared and analyzed. As shown in Figure 22a, the average task time costs are 8.15 s and 6.33 s when adopting the conventional model and the proposed system, respectively, and the average task time cost of the proposed system is reduced by 22.4% compared with the conventional model. As shown in Figure 22b, the average correlation instantaneous rates are 0.0563 and 0.0412 for the proposed system using the conventional model and the proposed system, respectively, and the average correlation instantaneous rate of the proposed system is significantly reduced compared to the conventional model. As shown in Figure 22c, the effective output ratios are 74.43% and 82.27% using the conventional model and the proposed system, respectively, and the effective output ratios of the proposed system are increased compared to the conventional model.

A dedicated analysis of the Online Task 1 data was conducted to quantitatively validate the accuracy of the complete instruction transmission pathway from EEG signals to exoskeleton execution. This analysis evaluated the fidelity of the entire control chain, spanning from decoding movement intention to physical actuation. The results demonstrate (1) an overall BCI command recognition accuracy of 80.4% across all subjects, confirming reliable decoding of movement intentions from EEG signals, and (2) an instruction execution success rate of 96.2% for correctly decoded commands, indicating that the exoskeleton consistently executed the intended movements when commands were properly interpreted. This high execution success rate, verified through synchronized exoskeleton joint sensor data and motion capture recordings, provides direct evidence of the system’s capability to accurately translate brain-derived commands into corresponding physical actions, thereby validating the effectiveness of the closed-loop control system.

In the Task 2 experiment, the task time cost, relevant instantaneous rate, and effective output ratio of the conventional model and the proposed system are compared and analyzed. As shown in Table 2, the average task time costs are 46.63 s and 37.94 s, the average relevant instantaneous rates are 0.0597 and 0.0428, and the effective output ratios are 74.81% and 82.17% using the conventional model and the proposed system, respectively.

The 40 subjects in Online Experimental Task 1 were randomly and equally divided into 8 groups (5 people/group), and the average accuracy of EEG signal recognition in each group is shown in Figure 23. Among the 8 groups of teams, the minimum value of the average accuracy was 78.025% and the maximum value was 83.462%, and the average accuracy of the 40 subjects reached 80.4%, which met the requirements for completing the experiment. While the groups’ average recognition accuracy was 80.4%, performance varied across individuals, with a standard deviation of ±4.2% (range: 72.5% to 87.5%). Similarly, the correlated instantaneous rate showed a standard deviation of ±0.0035 around the group mean. This variability is consistent with the well-documented phenomenon of ‘BCI illiteracy’ or performance variation in SSVEP-based paradigms.

To isolate the contribution of the improved FBCCA decoder from the AR-VS paradigm, an offline analysis was performed by re-processing the collected EEG data with the standard CCA method. The FBCCA decoder achieved a significantly higher average classification accuracy of 80.4% compared to 72.1% for the standard CCA decoder (*p* < 0.05), validating the individual contribution of the improved decoding algorithm to the overall system performance. To show the effect of using the brain-controlled hip exoskeleton more objectively, a questionnaire was administered to the 40 subjects after all the experiments were finished. The main content of the questionnaire included assessment indicators of satisfaction, maneuverability, adaptability, and stability, each of which was scored out of 10 points. The results of the questionnaire are shown in Figure 24. The results show that the brain-controlled hip exoskeleton system had an average satisfaction score of 8.9, maneuverability of 9.1, adaptability of 8.7, and stability of 8.8, which were significantly higher than those of the traditional model.

## 5. Discussion

Neurological disorders and traumatic injuries often lead to impaired hip motor function and persistent gait abnormalities during late-stage rehabilitation. While existing exoskeletons—including anthropomorphic and soft designs—commonly suffer from joint misalignment and direct pulling forces, this study introduces a wearable pneumatic hip exoskeleton that mimics the biological hip’s actuation characteristics to address these limitations. Experimental validation confirmed the structural efficacy of the proposed design: misalignment was minimized to 11.38 mm in Experiment 1, while force-tracking performance in Experiment 2 demonstrated that the assistive profiles closely matched reference trajectories during walking. The exoskeleton delivered maximum average forces of 18 N on level ground and 20.7 N on a 5° slope, indicating its capability to support gait training in patients with hip impairment. In terms of control, the integrated AR-VS paradigm elicited robust brain responses at target frequencies, enabling effective online BCI operation, though the use of a realistic scene as the visual background introduced minor interference with Steady-State Visual Evoked Potentials. During online testing, the system assisted users in completing 10 gait cycles with a mean task time of 37.94 s, an average instantaneous rate of 0.0428, and an effective output ratio of 82.17%. These results reflect significant improvements over the conventional model, with reductions of 18.64% in task time and 28.31% in instantaneous rate, alongside a 7.36% increase in output efficiency (all *p* < 0.05). The proposed system demonstrates potential for broader rehabilitation applications, and its proactive training paradigm may help enhance patient engagement and alleviate psychological distress.

It is important to note that this study serves as a proof-of-concept validation conducted on healthy subjects. While the results demonstrate the technical feasibility and advantages of the proposed system, the sample size of 40 healthy volunteers, though consistent with preliminary studies in the field [35,36], and the absence of clinical participants limit the direct generalizability of the findings to patient populations undergoing advanced rehabilitation. The primary aim at this stage was to establish a robust technical foundation by controlling for the confounding variables inherent in pathological conditions. Consequently, the promising results herein warrant further investigation in clinical settings.

This study presents a hip exoskeleton system driven by a hybrid EEG–AR-VS con-trol framework, which demonstrates substantial improvements across several key performance metrics relative to existing advanced systems (Table 3). In terms of con-trol methodology, the integration of electroencephalography-based intent recognition and augmented reality-based visual stimulation enabled a user calibration time of only 10 minutes, outperforming the 20 minutes required by the crank-angle-triggered hu-man-in-the-loop optimization strategy reported in Ref. [55]. Mechanically, the redun-dant self-aligning mechanism based on a parallel drive with flexible support units re-duced parasitic forces by 82%, significantly enhancing backdrivability and addressing the limitations of conventional rigid exoskeleton structures [40]. In functional assess-ment, the periodic segmented hybrid control strategy improved gait symmetry by 30%, a result that not only surpasses the bench-validated admittance control approach in Ref. [56] but also indicates more targeted neurorehabilitation benefits compared to the general improvements in gait and metabolic efficiency described in Ref. [57]. Further-more, multi-sensor fusion incorporating EEG, inertial measurement units, and force feedback contributed to robust motion intent decoding and adaptive control. Overall, while maintaining a responsive delay of 150 ms, the proposed system exhibits distinc-tive advantages in enhancing neural engagement and locomotor coordination, offering a promising technical pathway for the next generation of personalized rehabilitation exoskeletons.

Furthermore, following the approach of studies that effectively separate the evalua-tion of mechanical hardware from algorithmic contributions [58], our comparative and offline analysis helps to isolate and quantify the individual value of the key innovations in our system, namely the AR-VS paradigm and the FBCCA decoder.

This study has several limitations. First, the system requires users to have a baseline level of cognitive function and sufficient lower-limb stability, making it likely unsuitable for patients with severe cognitive deficits or in the flaccid stage of paralysis. Second, the model’s performance was validated only under controlled laboratory conditions. Its effi-cacy at gait speeds faster than the slow, constant speed tested remains unverified. Fur-thermore, the machine vision component depends on adequate lighting, and the mechan-ical system’s safety and performance were not tested on uneven terrain, slippery floors, or stairs.

In addition to these system-level shortcomings, several technical limitations persist. First, the kinematic model for misalignment analysis, while experimentally validated, does not capture the effects of soft tissue compliance and contact mechanics, which could be further explored using the Finite Element Method (FEM) or detailed multibody dynamics simulations in the future. Second, the inability to eliminate external signal interference and measurement errors in human gait remains a challenge. Among these, artifacts in the EEG signal have the greatest impact on the classification accuracy of exoskeleton control com-mands. Specifically, both EEG artifacts from physiological sources and environmental in-terference may affect system performance in real-world settings. Finally, the observed in-ter-subject variability in BCI performance, as demonstrated by our results, represents an-other important factor affecting system consistency. Moreover, the system’s efficacy and adaptability for clinical populations with diverse pathological characteristics require fur-ther investigation.

The observed inter-subject variability in accuracy and CIR necessitates consideration of factors that may predict BCI performance. While the scope of this study did not include a formal investigation of all potential physiological or psychological predictors, we noted that subjects who self-reported as regular players of video games (*n* ≈ 12) demonstrated a non-significant trend towards higher average accuracy (82.1%) compared to non-gamers (79.5%). This aligns with studies suggesting that prior experience with dynamic visual environments may enhance SSVEP responsiveness [59,60]. Future work should systematically collect subject characteristics such as age, gender, cognitive load, and detailed gaming history to build predictive models of performance for our specific AR-VS paradigm.

Future work will focus on transitioning the system from a laboratory prototype to clinical application through three key steps. First, we will validate safety and efficacy with target patient populations in controlled settings to establish clinical feasibility. Second, we will expand testing to real-world clinical environments to evaluate practical integration into therapeutic practice. Finally, we will pursue technological maturation through system miniaturization and multi-sensor fusion to enable future community- and home-based rehabilitation.

In summary, this work establishes a paradigm for integrated micro-systems in rehabilitation robotics, demonstrating the effective fusion of a BCI, embedded intelligence, and micro-actuation within a wearable platform.

## 6. Conclusions

In this study, we present a novel hip exoskeleton for gait rehabilitation and its accompanying AR-VS paradigm-based control system. The exoskeleton design integrates a flexible support unit with a parallel drive mechanism, utilizing kinematic redundancy to achieve self-alignment with the human hip joint and effectively minimize parasitic forces. The experimental results demonstrate that the system limits hip joint misalignment to 11.38 mm and ensures that assistive forces align with the intended direction of motion. The exoskeleton provides maximum average assistive forces of 18 N on level ground and 20.7 N on a 5° slope, meeting the requirements for late-stage rehabilitation training. Additionally, the brain-controlled system incorporating BMI and AR-VS paradigms, along with a gait phase-dependent hybrid controller, significantly enhances operational efficiency. Online experiments show that the system assisted subjects in completing 10 gait cycles with an average task time of 37.94 s, achieving an 18.64% reduction in completion time, a 28.31% decrease in relevant instantaneous rate, and a 7.36% improvement in effective output ratio compared to conventional approaches. These findings validate the system’s efficacy in enabling efficient and stable human–exoskeleton collaboration. The proposed framework offers valuable insights for future development of multi-joint exoskeleton control systems. Future studies will focus on clinical translation, with validation of system safety, usability, and therapeutic efficacy in patients with hip motor impairments, which is essential for optimizing the system for diverse rehabilitation needs.

## Figures and Tables

**Figure 1 micromachines-16-01364-f001:**
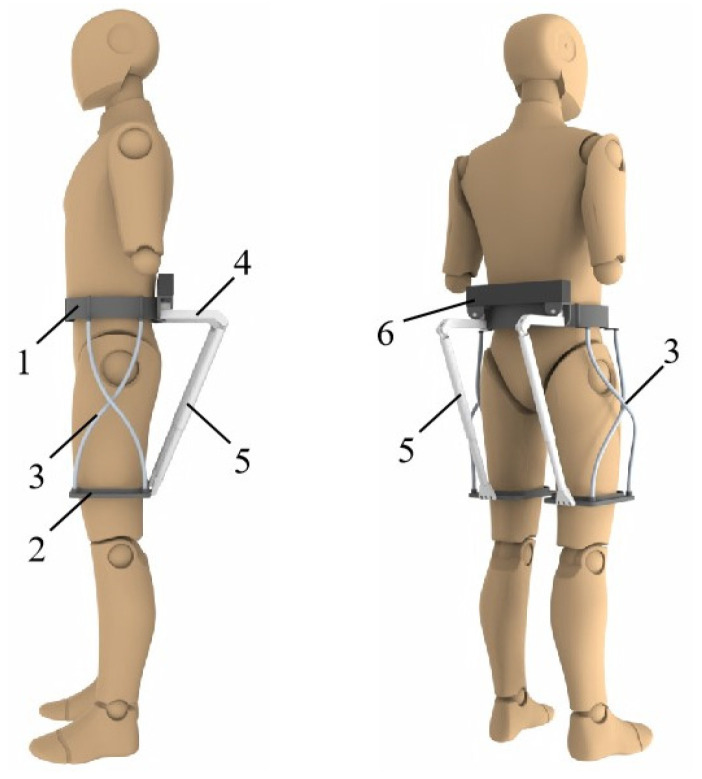
Three-dimensional model of the proposed self-aligning hip exoskeleton. The key components are labeled as follows: (1) waist belt, serving as the primary support interface; (2) leg brace, connecting to the user’s thigh; (3) flexible strut, enabling passive compliance to mitigate joint misalignment; (4) parallel mechanism, providing two degrees of freedom (flexion/extension and abduction/adduction) per leg; (5) drive unit, generating assistive torque; and (6) control unit, housing the embedded system. The design incorporates kinematic redundancy through the flexible support units to achieve alignment with the biological hip joint.

**Figure 2 micromachines-16-01364-f002:**
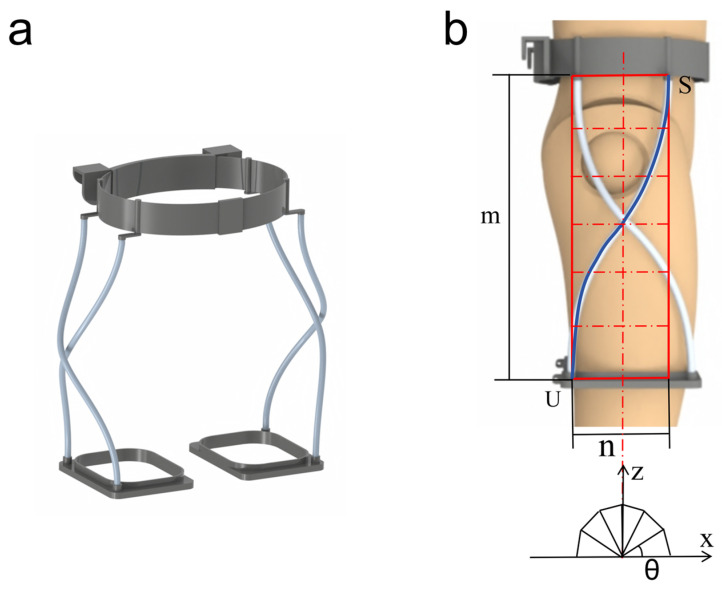
The flexible support unit of the hip exoskeleton. (**a**) Three-dimensional model showing the assembly of the waist belt, the leg braces, and the four twisted flexible struts. (**b**) Spatial projection of a single strut for the right leg, defining its geometrical parameters: the vertical projection length m and the horizontal projection length n. The S-side denotes the connection plane to the belt, while the U-side denotes the connection plane to the leg brace. This design allows the unit to adapt to positional deviations during movement.

**Figure 3 micromachines-16-01364-f003:**
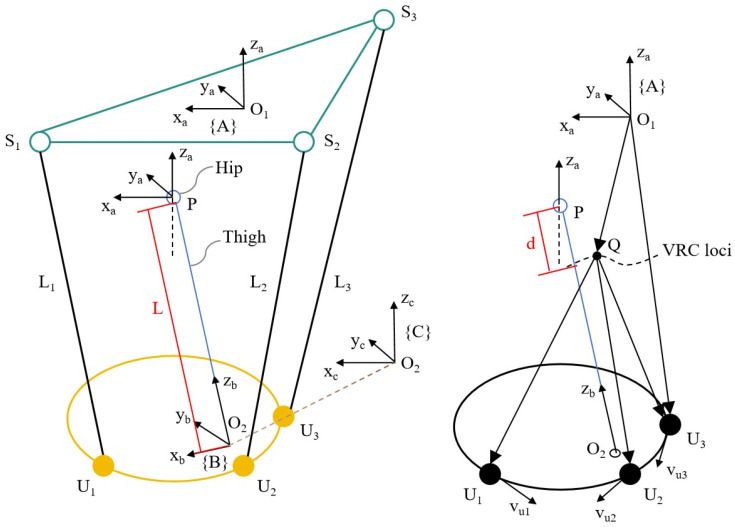
Geometric modeling of human–machine systems.

**Figure 4 micromachines-16-01364-f004:**
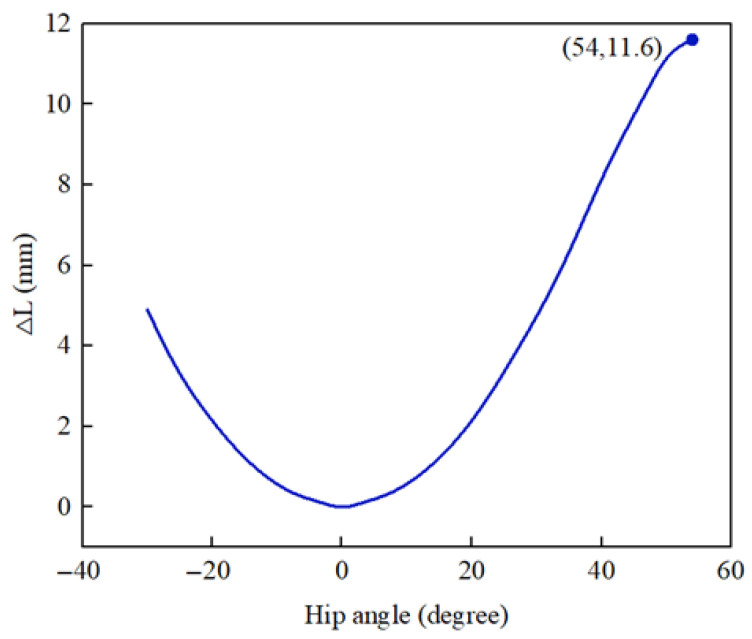
Calculation results of sliding distance ∆L.

**Figure 5 micromachines-16-01364-f005:**
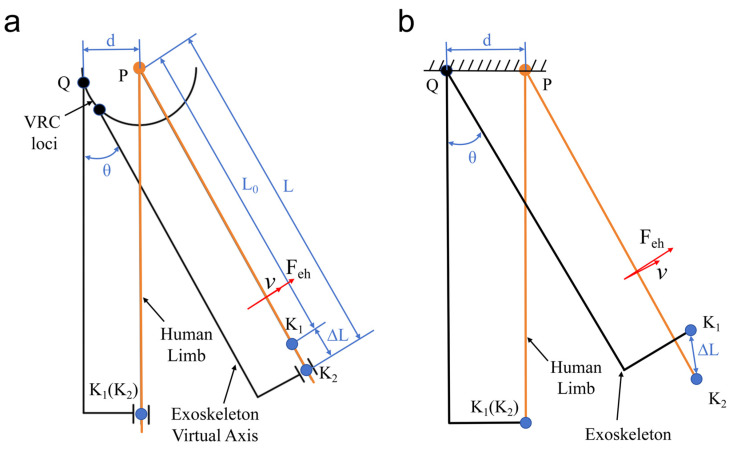
Plot of sliding distance ∆L versus misalignment d. (**a**) The designed hip exoskeleton. (**b**) Traditional rigid hip exoskeleton.

**Figure 6 micromachines-16-01364-f006:**
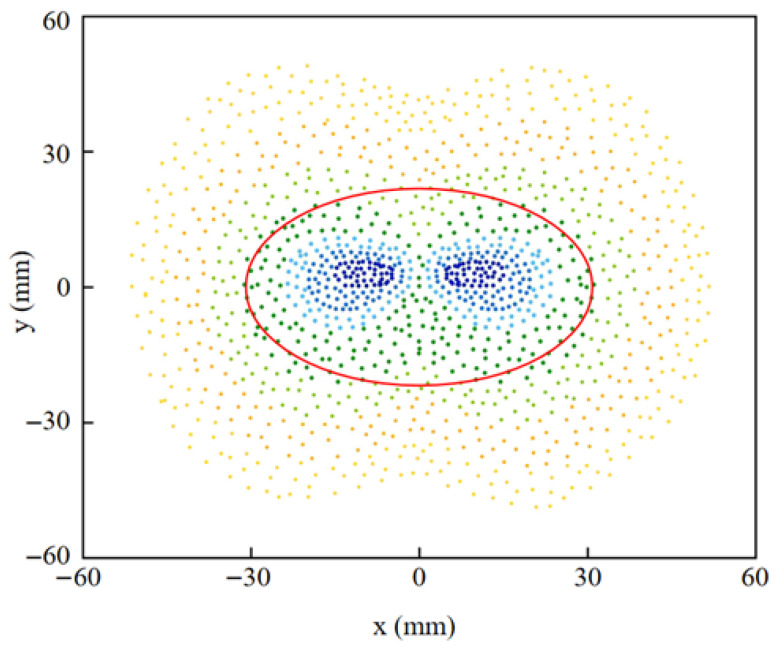
Distribution of misalignment d.

**Figure 7 micromachines-16-01364-f007:**
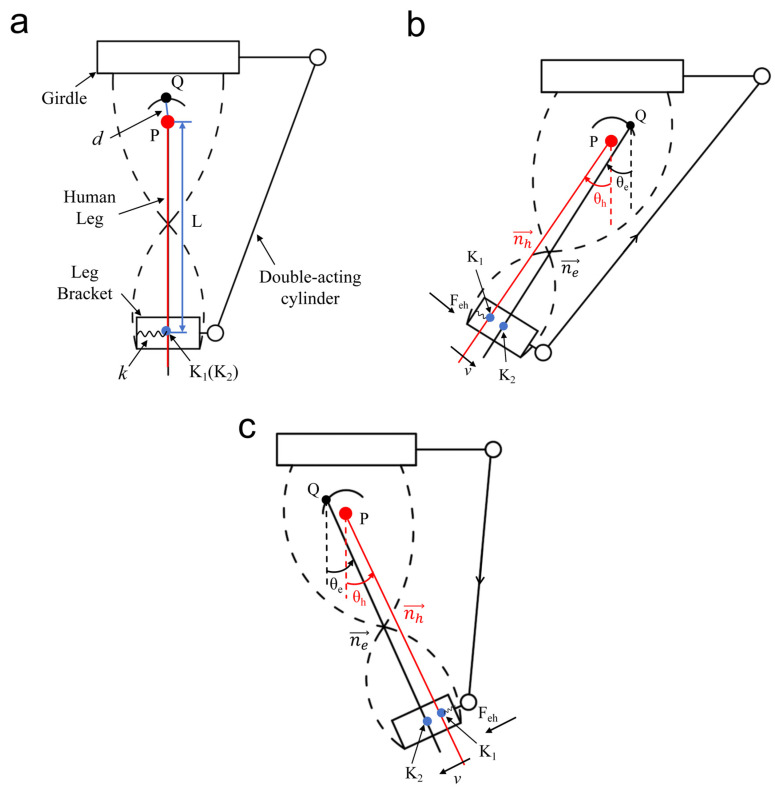
Interaction modeling of the human left leg with the hip exoskeleton. (**a**) Initial state of normal standing. (**b**) The exoskeleton assists in returning the thigh from a flexed state to a normal standing position. (**c**) The exoskeleton assists the thighs in returning from an extended state to a normal standing position. K_1_ and K_2_ are the contact points on the human leg and the leg brace, respectively.

**Figure 8 micromachines-16-01364-f008:**
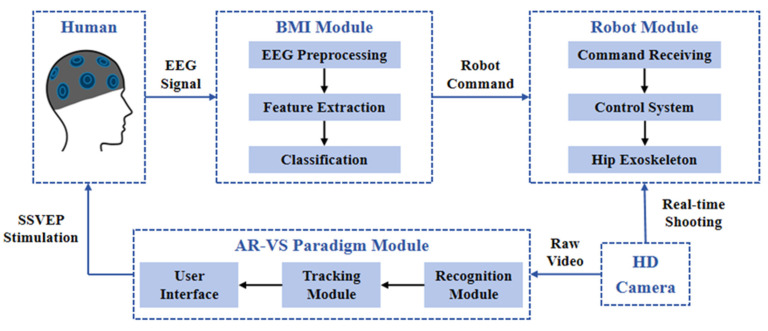
The systematic framework for the brain-controlled hip exoskeleton.

**Figure 9 micromachines-16-01364-f009:**
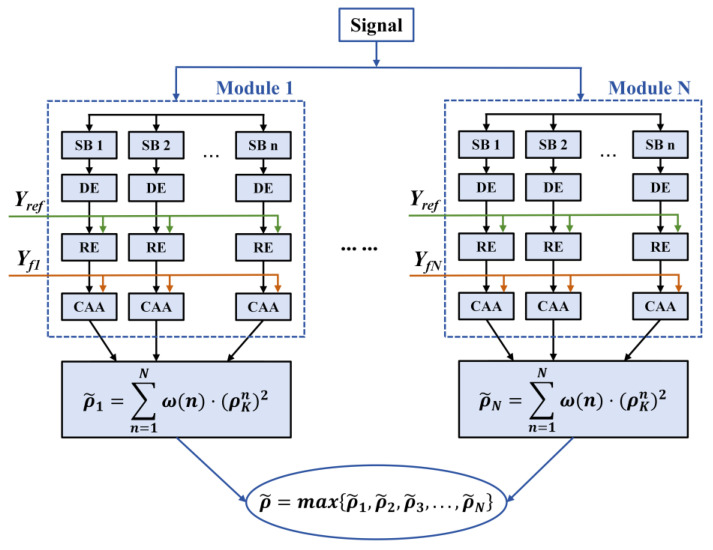
Decoding algorithm flowchart.

**Figure 10 micromachines-16-01364-f010:**
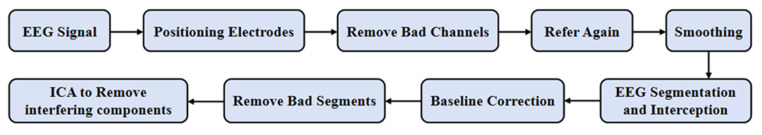
EEG signal preprocessing process.

**Figure 11 micromachines-16-01364-f011:**
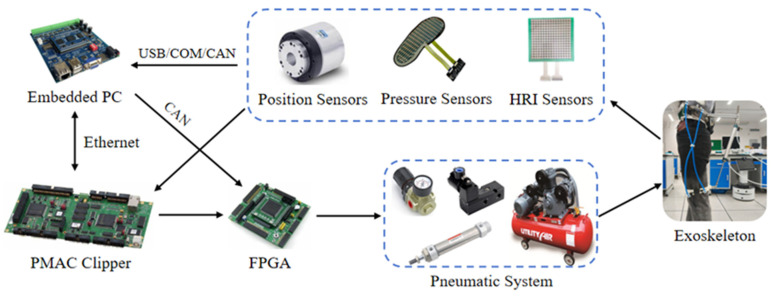
Hardware components of the robot control module.

**Figure 12 micromachines-16-01364-f012:**
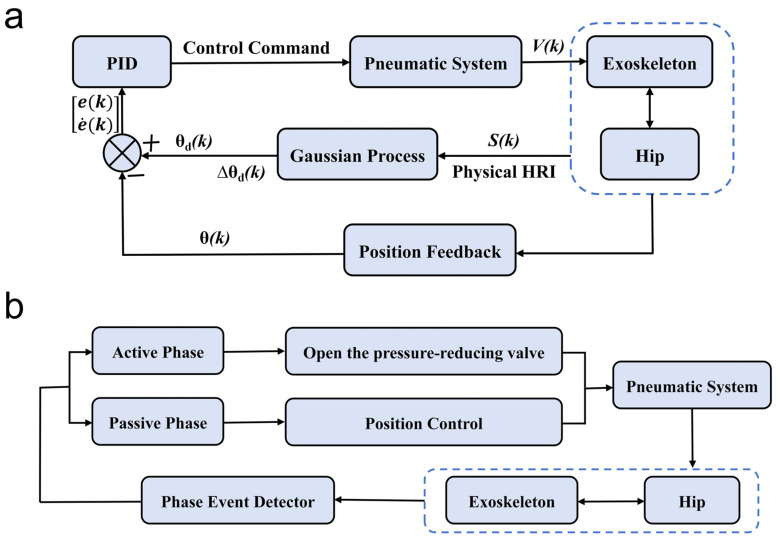
A hybrid control method based on gait cycle segmentation. (**a**) Passive phase position control strategy. (**b**) Active phase tracking control strategy.

**Figure 13 micromachines-16-01364-f013:**
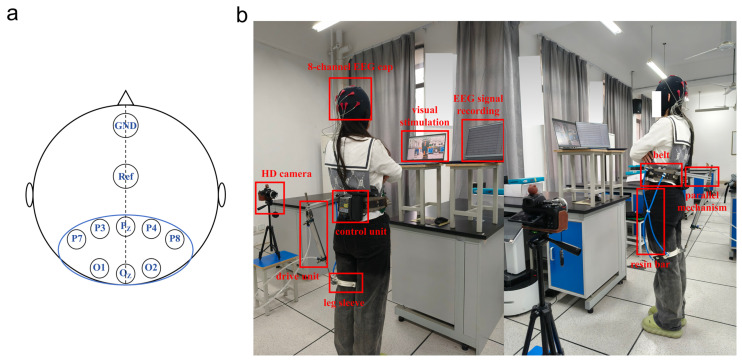
Experimental equipment. (**a**) Acquisition of EEG signals. (**b**) Experimental setup.

**Figure 14 micromachines-16-01364-f014:**
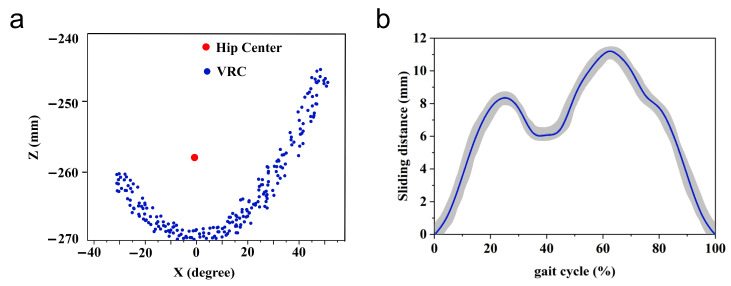
Verification experiment. (**a**) VRC instantaneous position mapped to the sagittal plane during leg motion. (**b**) Relative sliding distance between the leg braces and the thighs during one cycle of motion.

**Figure 15 micromachines-16-01364-f015:**
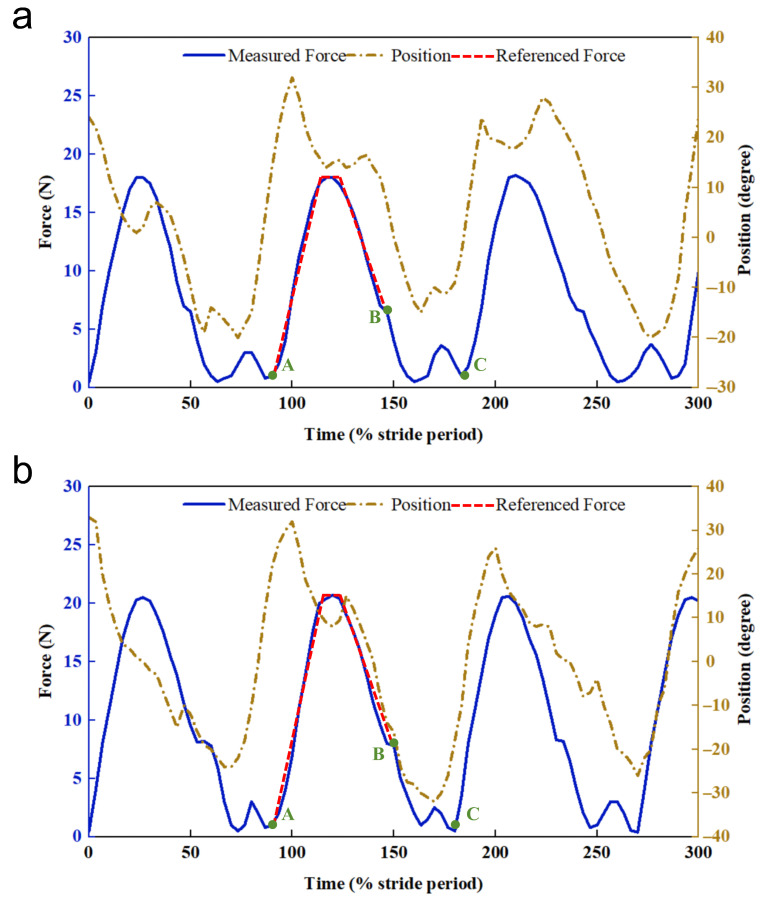
Force tracking of the hip exoskeleton over multiple consecutive gait cycles during assisted walking. (**a**) Walking at a constant speed on a horizontal surface. (**b**) Walking at a constant speed on a 5° sloped surface. The red line between points “A” and “B” in the figure is the expected reference force curve, adapted from the curves in the literature [53,54].

**Figure 16 micromachines-16-01364-f016:**
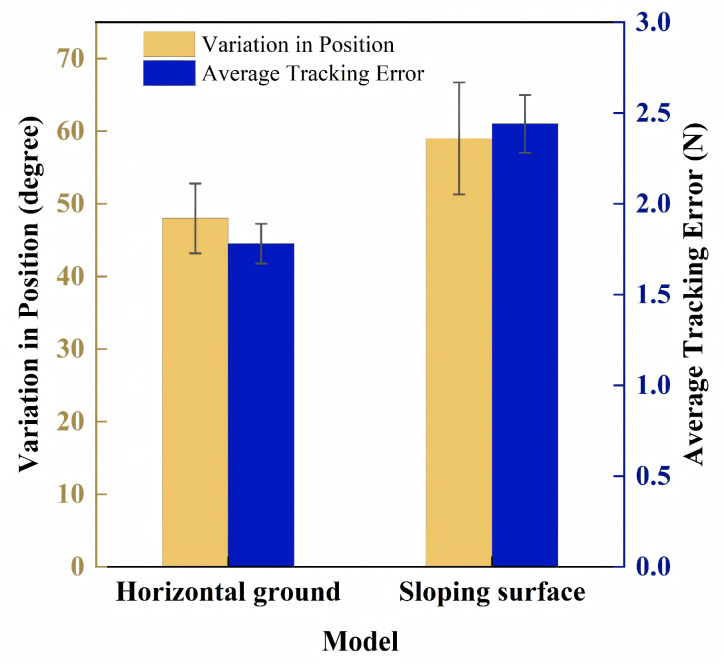
Amount of change in the angular position of exoskeleton joints with force-tracking error.

**Figure 17 micromachines-16-01364-f017:**
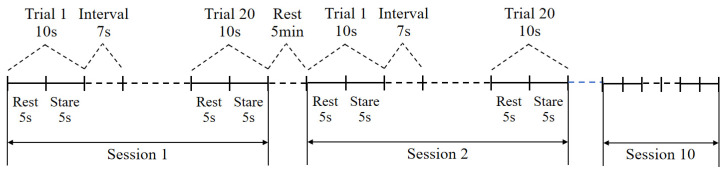
Offline experimental protocol in the time domain.

**Figure 18 micromachines-16-01364-f018:**
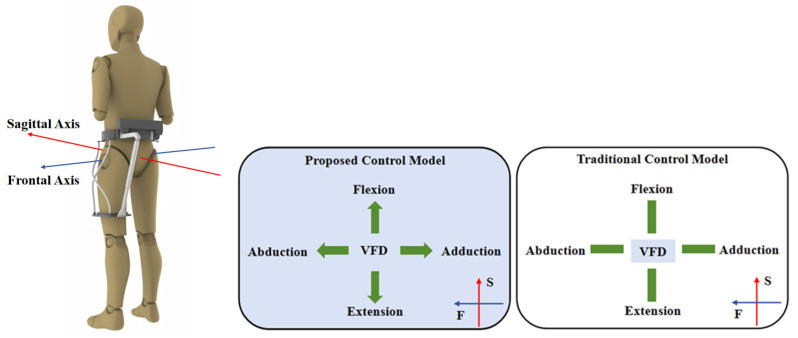
Flashing stimuli were moving in various directions and at different speeds.

**Figure 19 micromachines-16-01364-f019:**
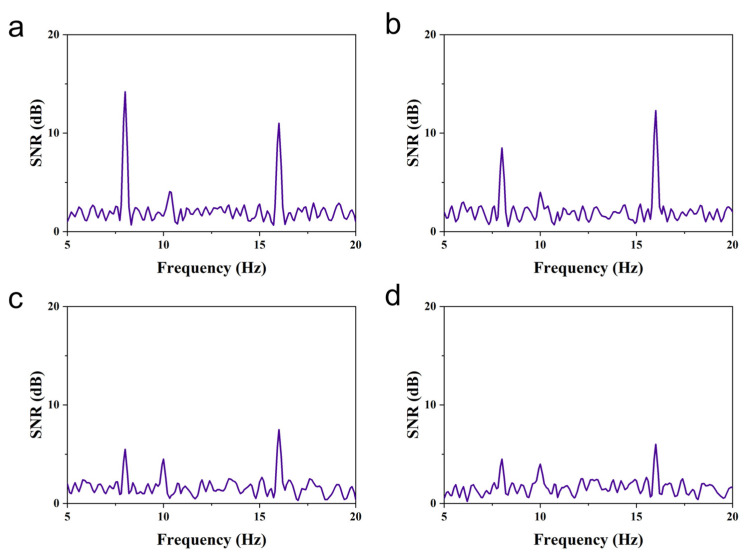
SNR of EEG signals induced by flicker stimulation in subjects with different movement speeds at 8.0 Hz. (**a**) A speed of 1 cm/s. (**b**) A speed of 3 cm/s. (**c**) A speed of 5 cm/s. (**d**) A speed of 7 cm/s.

**Figure 20 micromachines-16-01364-f020:**
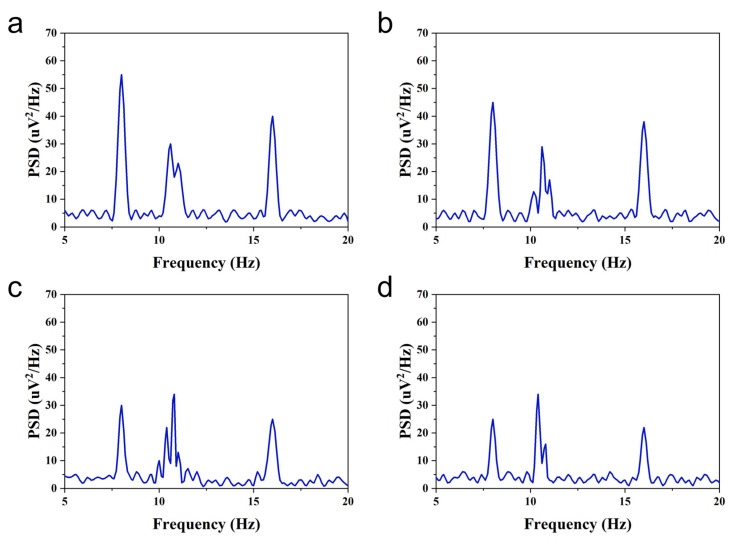
PSD of EEG signals induced by flicker stimulation in subjects with different movement speeds at 8.0 Hz. (**a**) A speed of 1 cm/s. (**b**) A speed of 3 cm/s. (**c**) A speed of 5 cm/s. (**d**) A speed of 7 cm/s.

**Figure 21 micromachines-16-01364-f021:**
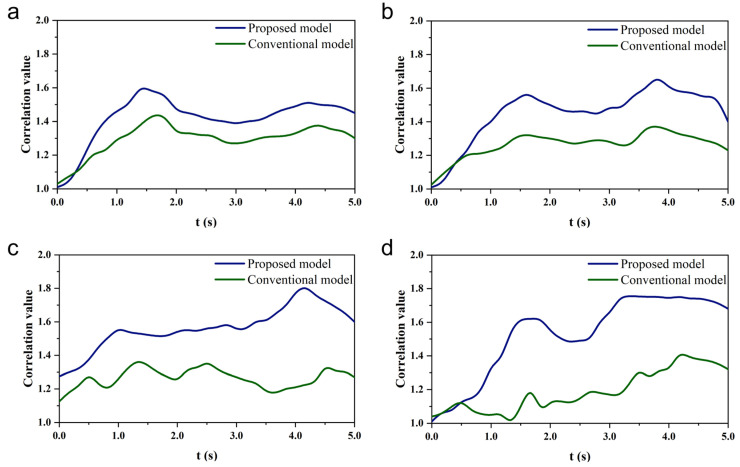
Changes in correlation values during controlled hip movements by one subject in online experimental Task 1. (**a**) Hip extension (7.2 Hz). (**b**) Hip flexion (7.6 Hz). (**c**) Hip abduction (8.0 Hz). (**d**) Hip adduction (8.4 Hz).

**Figure 22 micromachines-16-01364-f022:**
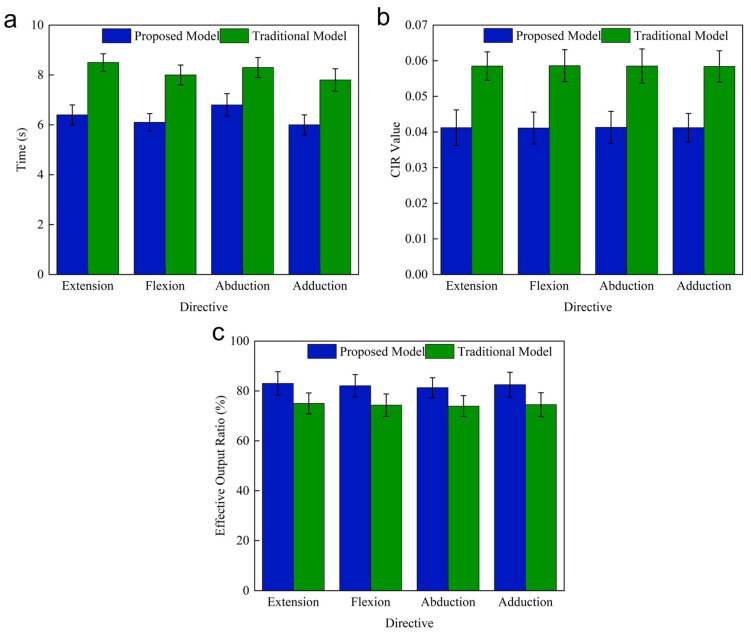
Comparison of the performance of the conventional model and the proposed system in online experiment Task 1. (**a**) Task time cost. (**b**) Correlated instantaneous rate. (**c**) Effective output ratio.

**Figure 23 micromachines-16-01364-f023:**
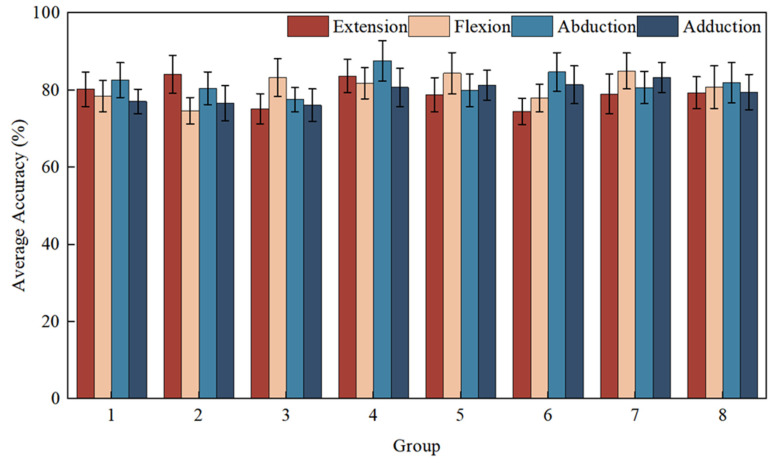
Mean accuracy of EEG signal recognition for 8 groups of subjects in Online Experimental Task 1.

**Figure 24 micromachines-16-01364-f024:**
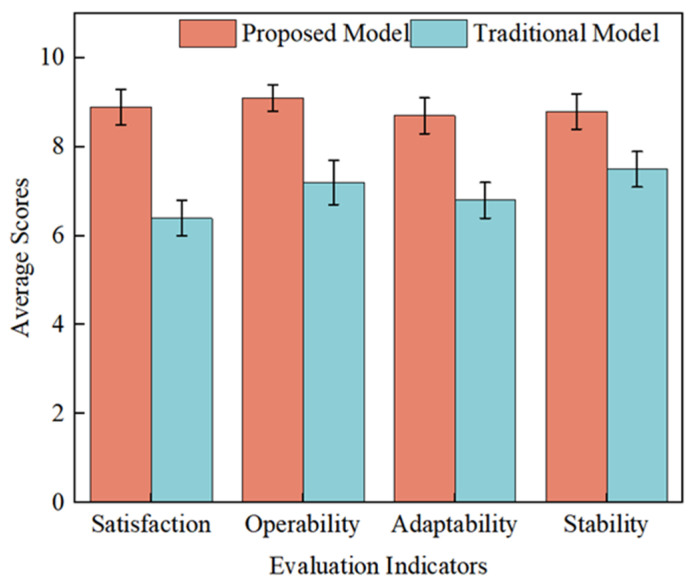
Results of the questionnaire.

**Table 1 micromachines-16-01364-t001:** Hip exoskeletons and the angular displacement of human biological hip joints.

Hip	Exoskeleton Max.	Human Max.	Human Walking Max.
Flexion/Extension	54°/−54°	≥54°/30°	32°/22°
Abduction/Adduction	57°/−35°	54°/30°	8°/6.5°

**Table 2 micromachines-16-01364-t002:** Comparison of the performance of the conventional model and the proposed system in Online Experimental Task 2.

	Model	Proposed Model	Conventional Model
Parameters			
Time		37.94 s	46.63 s
CIR value		0.0428	0.0597
Effective output ratio		82.17%	74.81%

**Table 3 micromachines-16-01364-t003:** Comparison of the proposed EEG-controlled hip exoskeleton with advanced systems.

Comparison Terms	This Study	Literature [55]	Literature [40]	Literature [56]	Literature [57]
Control methods	EEG + AR-VS paradigm + hybrid control	Crank-angle-triggered + HITL optimization	Quasi-stiffness Impedance control	Admittance control	Delayed Output Feedback Control
Driver design	Parallel drive + flexible support units	Parallel quasi-direct drive	Parallel quasi-direct drive	Single-actuator 6-bar Stephenson-III mechanism	Single actuator per hip
Kinematic properties	Redundant self-aligning design (parasitic force decrease 82%)	High backdrivability for low resistance	High backdrivability and trans-parency	3-DOF spherical motion with 1-DOF actuator	Fixed structure with mechanical stoppers
Gait-assisted strategies	Periodic segmented hybrid control	Phase-based torque profile (Extension/Flexion)	Task-specific impedance (Walking and Lifting)	Frontal and sagittal plane assistance	Assistance/Resistance torque at hip joint in sagittal plane
Sensor fusion	EEG + IMU + force feedback	Motion capture + Motor encoder	IMU + Encoder	Torque sensor + IMU + Strain gauges (3-axis)	IMU + Hip joint angle/velocity sensors
Experimental effect	Gait symmetry increase 30%	Net metabolic cost decrease 31.4%	Metabolic cost decrease 16.7% (Lifting), decrease 19.4% (In-cline walk)	Bench validation of moment direction tracking	Gait speed, balance, muscle strength, metabolic efficiency increase
User adaptability	AR calibration (10 min)	Human-in-the-loop optimization (20 min)	Manual task selection and preference tuning	Multi-objective optimization (NSGA-II)	Therapist-controlled settings

## Data Availability

The original data presented in the study are openly available at https://github.com/B-Cheng26/Data (accessed on 26 November 2025). Any further data are available from the corresponding author on request.

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
