# Peer review of "Design and Validation of a Brain-Controlled Hip Exoskeleton for Assisted Gait Rehabilitation Training"

_micromachines, 2025, doi:10.3390/mi16121364_

Round 1
Reviewer 1 Report
Comments and Suggestions for Authors
The paper presents a novel brain-controlled hip exoskeleton system that integrates a kinematically redundant flexible mechanism, an augmented reality-visual stimulation (AR-VS) paradigm, and a gait phase-dependent hybrid controller for gait rehabilitation. The study addresses a significant clinical need, and the integration of multiple advanced technologies is commendable. The experimental design is comprehensive, involving both offline and online validation with healthy subjects, and the results demonstrate promising performance improvements over traditional methods. However, several issues require attention to enhance the manuscript's clarity, rigor, and reproducibility before it can be accepted for publication.
- The manuscript lacks sufficient detail regarding key hardware components and experimental parameters. For instance, the specific models of the EEG cap, electrodes, inertial sensors, digital displacement sensor, and pneumatic components (e.g., cylinder bore size, spring constant) are not provided. The exact filter settings (e.g., band-pass ranges beyond 0.1-30 Hz) and the specific machine vision algorithms used for hip tracking are also omitted. Including these details is crucial for experimental reproducibility.
- The results are promising, showing reductions in task time and misalignment. However, the discussion regarding the system's operational boundaries and limitations remains somewhat superficial. The authors should more explicitly define the system's limitations concerning gait speed ranges, the severity of patient impairment it can accommodate, and the impact of different environmental factors (e.g., lighting conditions for the AR-VS module, floor surfaces). Future work directions, such as testing with actual patient populations and in real-world clinical settings, should be more concretely outlined.
Some specific comments requiring attention:
- Several figures are complex and could be improved for readability. Figure 9 (FBCCA flowchart) is particularly crowded. Consider simplifying the layout or splitting it into multiple sub-figures. Figures 19 and 20, which contain multiple subplots, would benefit from clearer labels and a more organized arrangement.
- Ensure all figure captions are consistently formatted, centered, and clearly separated from the figures. Some captions (e.g., Fig. 1, Fig. 2) are brief and could be expanded to include more contextual information.
- While most major abbreviations are defined, some are missed or defined late. Please ensure every abbreviation is defined upon its first use in the main text. A nomenclature table would be highly beneficial.
- Some terms are used inconsistently, such as "flexible support unit" vs. "flexible strut," and "leg brace" vs. "leg holder." The manuscript should use standardized terminology throughout.
- While safety measures for SSVEP stimulation are mentioned, more details on ethical approval, participant consent procedures, and specific safety protocols (e.g., luminance levels, emergency stop implementation) would strengthen the methodology section.
- The limitations section is brief. Expanding on the impact of EEG artifacts, environmental interference, and the system’s performance with clinical populations would provide a more balanced perspective.
Author Response
Responses to Reviewers' Comments
We sincerely thank the reviewers for their insightful comments and constructive suggestions, which have significantly helped us improve the quality of our manuscript. We have carefully addressed all the points raised. The revisions in the manuscript are marked in red for the reviewers' convenience. Our point-by-point responses are detailed below.

Reviewer 2 Report
Comments and Suggestions for Authors
This study presents a novel wearable hip robotic system controlled by an active brain-computer interface-driven system. Parasitic force can be minimized through a flexible support unit and a parallel drive mechanism. Augmented reality and visual stimulation are integrated to enhance motor intent recognition.
Although the paper presents a highly original contribution to the design of wearable hip rehabilitation robots, some comments may help the authors improve their manuscript.

Author Response

(The authors gave the same response as above.)

Reviewer 3 Report
Comments and Suggestions for Authors
The manuscript describes a wearable hip exoskeleton that uses a flexible support unit with a parallel drive, combined with an AR-based SSVEP paradigm and a gait phase–dependent hybrid controller. The system is evaluated on 40 healthy subjects and shows improved task time, correlated instantaneous rate, and effective output ratio compared to a conventional control mode. The mechanical modelling, AR-VS concept and the integration with an FBCCA-based decoder and Gaussian process for gait prediction are interesting and potentially valuable.
1. The experiments were conducted on 40 healthy volunteers, too small a number to justify its applicability to advanced rehabilitation. Did the authors conduct further tests?
2. The chosen AR-VS SSVEP frequencies (7.2, 7.6, 8.0, 8.4 Hz) are close to the low-frequency band where SSVEP responses and alpha rhythms can interact. How were these frequencies selected, and did you test for frequency contamination or recognition reduction due to alpha activity?
3. The FBCCA decomposition parameters and the weighting function f(n) are critical to performance. How were the constants a and b in f(n) selected, and were they cross-validated? Please provide the hyperparameter tuning procedure and sensitivity analysis.
4. The Gaussian process model is used for passive phase prediction. Which kernel was used? How was the kernel hyperparameter optimisation performed? How does the GP performance compare to simple linear regression or an LSTM on the same data?
5. How robust are the mechanical misalignment and slip measurements with respect to marker placement and motion capture calibration?
6. Online experiments use a “conventional model” as a benchmark. Authors should precisely define the hardware, stimuli, and decoder settings, justifying their choice. Furthermore, could an ablation, such as removing AR tracking or replacing FBCCA with standard CCA, better isolate contributions? The authors should supplement the Methods section, which discusses the experimental protocol, with the study doi: 10.3390/computers14090344 because it demonstrates the value of ablation studies that separate FEM/mechanical contributions from AI/algorithmic contributions.
7. SSVEP stimuli can trigger photosensitive responses in susceptible individuals. Screening and luminance limits are listed; can the authors report whether any adverse events occurred during/after the sessions and whether any subjects experienced discomfort or transient symptoms? The authors should also report the number of emergency stop activations (if any).
8. The article provides group averages for accuracy (mean 80.4%) and CIR. Should the authors provide variability per subject and discuss differences between subjects? Are there subject characteristics that predict performance?
9. Should the authors provide more detail on pneumatic control latency, PMAC/FPGA loop times, and whether the 50 ms control cycle meets real-time requirements for gait? Should the authors report the worst-case actuator response time?
10. The authors should report the average force tracking (18 N on flat ground, 20.7 N on slopes) and provide a justification that these force levels are clinically significant for advanced hip rehabilitation and explain the targeted assistance profile during the gait cycle.
11. The modelling section uses kinematic derivations for misalignment. Have these been validated against FEM or multibody dynamics simulations to capture tissue compliance and soft contact effects?
Author Response

(The authors gave the same response as above.)

Round 2
Reviewer 3 Report
Comments and Suggestions for Authors
I thank the authors for their replies to the comments. I have no further comments to make.